# A novel mitovirus associated with the fungal entomopathogen *Zoophthora radicans*

**Meaghan J. Adler**[¤]*, **McKayla M. Martin, Paula Rozo-Lopez, Benjamin J. Parker**[¤]*

Department of Microbiology, University of Tennessee, Knoxville, Tennessee, United States of America

¤ Current Address: Department of Biology, University of North Carolina at Chapel Hill, Chapel Hill, North Carolina, USA
* meaghan.adler@gmail.com (MJA); bjp@utk.edu (BJP)

## Abstract

Metatranscriptome sequencing has emerged as a powerful tool for uncovering viral diversity in insects and their associated microbes. To explore viruses linked to the pea aphid (*Acyrthosiphon pisum*), we performed metatranscriptome sequencing on field-collected samples. In addition to several known plant viruses, we assembled the genome of a new virus homologous to species in the family *Mitoviridae*, which are positive-sense single-stranded RNA viruses that encode only an RNA-dependent RNA polymerase and typically replicate in mitochondria. Given the frequent association of mitoviruses with fungi and the presence of entomopathogenic fungal reads in our data-set, we conducted 18S amplicon sequencing and PCR screening on individual aphids. These analyses identified two fungal entomopathogens and uncovered an association between the mitovirus and *Zoophthora radicans*, a generalist aphid pathogen. Our findings shed light on the hidden microbial diversity in natural pea aphid populations and emphasize the utility of metatranscriptomics in identifying novel virus-host associations.

## Introduction

Metatranscriptome sequencing has become a critical tool for uncovering viral diversity, revealing viruses that were previously undetectable using traditional approaches [1,2]. Historically, most of what we know about insect-associated viruses is focused on microbes that are recognized pathogens or are transmitted as vectors of disease. Over the past two decades, however, metatranscriptomic studies have dramatically expanded our understanding of viral diversity, identifying numerous novel viruses associated with insects and other eukaryotes [3,4]. This technique, which involves sequencing RNA while often enriching for viral reads or depleting ribosomal RNA, is particularly valuable in field-based studies, where diverse viral communities can be detected in wild-collected samples [5]. A key challenge in these studies is determining the true host of newly identified viruses—whether they are infecting the insect itself or instead associated with another organism interacting with the insect.

**Data availability statement:** All meta-transcriptome and fungal 18s sequencing raw reads are available from NCBI under BioProject IDs, PRJNA1246236 (accession numbers SRX28243900- SRX28243901) and PRJNA1306634 (accession numbers SAMN50642901-SAMN50642919), respectively. Polerovirus genomes are available at NCBI accession: PV560925 and NCBI accession: PQ553207, Luteovirus genome is available at NCBI accession: PV611337. Zoophthora radicans mitovirus 1 is available at GenBank: PQ606655.1. https://www.ncbi.nlm.nih.gov/bioproject/1246236; http://www.ncbi.nlm.nih.gov/bioproject/1306634; https://www.ncbi.nlm.nih.gov/nuccore/PV560925.1/; https://www.ncbi.nlm.nih.gov/nuccore/PQ553207.1/; https://www.ncbi.nlm.nih.gov/nuccore/PV611337.1/; https://www.ncbi.nlm.nih.gov/nuccore/PQ606655.1.

**Funding:** This work was funded by National Science Foundation (NSF) Grant IOS-2152954 to BJP and by Grant DEB-2305653 to PRL. MJA was supported by the SMART scholarship funded by OUSD/R&E (The Under Secretary of Defense-Research and Engineering), National Defense Education Program (NDEP)/ BA-1, Basic Research. MMM was supported by a Microbiology Summer Research Award from the Department of Microbiology at UTK. A portion of this project was carried out by MMM in association with a course-based undergraduate research experience (CURE), which was funded by the College of Arts and Sciences at the University of Tennessee Knoxville. BJP is a Pew Scholar in the Biomedical Sciences, funded by the Pew Charitable Trusts. The funders had no role in study design, data collection and analysis, decision to publish, or preparation of the manuscript.

**Competing interests:** The authors have declared that no competing interests exist.

Pea aphids (*Acyrthosiphon pisum*) are an important model for studying host-microbe interactions [6]. Research has focused on the associations between pea aphids and heritable bacterial symbionts, which influence aphid interactions with host plants and natural enemies [7]. Like many species, *A. pisum* transmits various plant viruses [8], and individual viral taxa have been discovered in laboratory colonies of *A. pisum* [9–11]. However, the broader viral diversity associated with pea aphids, particularly in natural populations, remains unexplored.

Fungal entomopathogens also play a significant role in aphid ecology, particularly those in the subphylum *Entomophthoromycotina* [12,13]. This group includes both generalists (e.g. *Zoophthora radicans* and *Batkoa major)* and specialist pathogens (e.g. *Pandora neoaphidis)* that differ in their host ranges and infection strategies [14]. Mycoviruses associated with fungi represent a potentially important aspect of fungal biology, and early study uncovered mycoviruses associated with key fungal species in the *Entomophtoromycotina* [15]. For example, eight mitoviruses—a unique group of RNA viruses that replicate in mitochondria and encode only a viral RNA-dependent RNA polymerase—were identified within *Entomophthora muscae* [16]. Identifying and describing new fungal-virus relationships will provide insight into the evolution of these unique symbioses and their role in shaping fungal evolution and phenotypes.

In this study, we performed metatranscriptome sequencing on field-collected samples of pea aphids from East Tennessee, USA. Our analyses revealed several viral genomes, including a new species of *Mitovirus*. Because of the known association between mitoviruses and entomopathogens [16], we also performed 18S sequencing of wild-collected aphids and identified infections by two entomopathogenic fungi: *Pandora neoaphidis*, which was associated with visible fungal cadavers, and *Zoophthora radicans*, which was detected in asymptomatic individuals. PCR screening further demonstrated that the newly described *Mitovirus* was exclusively present in samples infected with *Z. radicans*, suggesting that this fungus serves as the viral host. Together, our findings highlight previously unrecognized viral diversity within aphid-associated fungi and provide new insights into the complex microbial interactions shaping aphid ecology.

## Methods

### Aphid collection

We collected adult pea aphids from species of host plants in the genus *Trifolium* (*T. repens* and *T. pratense*) and the genus *Vicia* in May 2023 at The University of Tennessee Research Park at Cherokee Farm. Samples were a mix of winged (alate) and unwinged (aptera) adults (metadata in S1 Table). We reared each aphid individually in a Petri dish on its original plant material with the stems embedded in 2% agar for 4 days, recorded whether aphids developed a visible fungal, bacterial, or parasitoid wasp infection, and then froze aphids individually in Eppendorf tubes at −80°C.

### RNA extractions

For a subset of aphids (see S1 Table), we extracted RNA from individual adults by adding 220µl of Bender Buffer to each tube and grinding using a sterile pestle. We

added 5μl of Proteinase K to each tube and incubated at 65°C for two hours. We then added 28μl of 8M KoAc to each tube and vortexed briefly. We placed tubes on ice and incubated them overnight at 4°C. The following day, we centrifuged the samples for 15 min at 20,000 RCF at room temperature and transferred 70μl of the supernatant into two separate Eppendorf tubes. One tube was used for RNA extraction and the other for DNA extraction, described below. To obtain total RNA, we added 300μl of Trizol to the 70μl supernatant, followed by 100μl of BCP, and mixed by hand for 20 seconds. We then allowed the samples to incubate at room temperature for 5 min and centrifuged at 4°C for 15 min at 12,000 RCF. We then transferred the clear, aqueous phase to a new Eppendorf tube. We precipitated the RNA with 150μl of 100% cold isopropanol and allowed this to incubate at room temperature for 10 min after briefly mixing the tube by inversion. Following incubation, we centrifuged the samples at 4°C for 10 min at 12,000 RCF and removed the supernatant. We performed two ethanol washes of the pellet by adding 300μl of 75% ethanol, followed by centrifugation at 4°C for 5 min at 7,500 RCF. We removed the ethanol and allowed the RNA pellet to air dry for 10 min. We resuspended the pellet in 40μl of warm, nuclease-free water and incubated the tube at 56°C for 5 min. We used a Nanodrop to record the quality and quantity of the extraction. We pooled RNA (equally by mass) into two pools of 5 aphid RNA extractions each for sequencing library and stored the samples at −80°C.

### Metatranscriptome sequencing and data deposition.

We performed metatranscriptome sequencing at Novogene Corporation Library preparation was conducted using ribosomal RNA depletion using the Illumina TruSeq Stranded Total RNA with Ribo-Zero Plus and NEBNext rRNA Depletion Kit. Each library was sequenced to ~9 billion base pairs with 150 bp paired-end reads on an Illumina Nova-Seq platform. Raw reads were deposited to NCBI, with BioProject ID PRJNA1246236 and accession numbers SRX28243900-SRX28243901.

### Virome characterization using CZID.org.

We used CZ ID Illumina mNGS metagenomic pipeline v.8.2, an online open-access platform for characterizing viral communities associated with metatranscriptome data [17]. Briefly, this pipeline includes adapter trimming and quality filtering; host reads were removed using the pea aphid reference genome [18] via alignment using STAR. Trimmomatic was used to remove adapter sequences, and PriceSeqFilter was used to remove low-quality reads. Mini-map2 [19] and Diamond [20] were used to align the remaining reads to the NCBI NT and NR databases. In parallel, SPADES [21] was used to de novo assemble short reads, and bowtie2 [22] was used to map reads back to the assembled contigs. We used threshold criteria of 10 reads per million (rPM) for viruses aligned to the NCBI NR database, and 1,000 rPM for bacteria and eukaryotes. With completion of the CZ ID pipeline, we de novo assembled reads with homology to each putative virus using Geneious Prime version 2025.0 using default parameters for the De Novo Assemble tool. With the assembled genomes, we used the NCBI Open Reading Frame Finder (https://www.ncbi.nlm.nih.gov/orffinder/, last modified 2021-05-04) to identify protein-coding regions of the viral genome (using genetic code 1 for putative viruses, and genetic code 3 Yeast Mitochondrial for the putative mitovirus). We used interpro (https://www.ebi.ac.uk/interpro/ release 103.0) to look for conserved genes to annotate viral genomes.

### *Mitovirus* PCR screening of field samples.

We designed primers to screen field samples for the mitovirus (mito_1751_F: CTAGGGTGAGGTCGCAGAT and mito_2251_R: GCGACGCTCTGACAGAATT). We extracted RNA from field-collected aphids and constructed cDNA via iScript cDNA Synthesis Kit by BIO-RAD, as directed by the manufacturer. We ran 25μL reactions with the forward and reverse primers at 320 nM (0.8 μL each), along with NEBNext High-Fidelity 2X PCR Master Mix and 40ng of cDNA. We included RNA from the original RNAseq study (pool 2) from which we obtained the *Mitovirus* reads (above) as a positive control. The PCR protocol included an initial step of 95°C for 30s, followed by 36 cycles of 95°C for 30s, 59.1°C for 30s, and 72°C for 30s, and a final elongation step of 72°C for 5 min.

### gDNA extraction.

We extracted genomic DNA from individual aphids using 70μl of the supernatant of the post-protein precipitation supernatant from the RNA extraction described above. We added 200μL of cold 100% EtOH to each sample, vortexed briefly, and incubated at −20°C for one hour. We then centrifuged the samples for 15 min at 20,000 RCF to pellet the DNA. We removed the supernatant, and washed with 200μL of cold 70% EtOH, vortexed briefly, and centrifuged for 5 min. We washed the pellet again with cold 100% EtOH, vortexed, and centrifuged for 5 min, and then air dried the pellet for 15 min. We resuspended the pellet in 21μl of nuclease-free water by incubating at 55°C for 10 min, briefly vortexed, and stored the samples at 4°C.

### Fungal 18S sequencing

Field samples were selected to include representatives of cadaverized and noncadaverized aphids, and mitovirus positive and negative aphids, with priority given to field samples with the highest quality DNA and RNA post-extraction. Amplicon sequencing of the fungal 18s V4 region was conducted at Novogene Corporation using the primers 528F (GCGGTAAT-TCCAGCTCCAA) and 706R (AATCCRAGAATTTCACCTCT). Libraries were sequenced on an Illumina NovaSeq 6000 to generate 250 bp paired-end reads, for a total of ~ 358 million base pairs across 19 samples. We processed amplicon reads using dada2 (version 1.26.0) with the following steps and the associated functions: filtering via filterandtrim, read error rates via learnError, dereplication via derepFastq, sample inference via dada with pseudo-pooling of all samples, and merged paired reads via mergePairs [23]. We generated an ASV table from the resulting merged fastq files using makeSequenceTable, and we removed chimeras with removeBimeraDenovo. We assigned taxonomy to each ASV with assignTaxonomy with a minimum bootstrapping confidence of 80 and using databases: silva_nr99_v138.1 with the species training set, and silva_132_18s.99 rep set [24,25]. We visualized the amplicon data using phyloseq [26]. We performed a fisher's exact test on the association between *Z. radicans* and positive bands for the mitovirus using fisher.test in R v.4.4.1. Raw reads were deposited to NCBI SRA with BioProject ID: PRJNA1306634.

## Results

### Metatranscriptome microbial composition

Taxonomic analysis of microbial reads in field pea aphid metatranscriptomes revealed a diverse community of microbes, including reads from aphid bacterial symbionts and from potential gut or surface bacteria known to be associated with aphids (Fig 1A). Pool 2 had a large number of reads associated with hymenopteran parasitoid wasps, suggesting that one or more of the aphids in this pooled sample were infected with a wasp egg or larvae (Fig 1A). Pool 2 also had reads from a fungal entomopathogen that was annotated by the CZ ID pipeline as *Pandora kondoiensis*, a fungal pathogen of aphids [27]. In both pools there were a significant number of viral reads, including a *Polerovirus* (*White clover mottle virus*; WCMV) and a *Luteovirus* (Soybean dwarf virus; SbDV). In pool 2 there were also reads with homology to viruses in the family *Mitoviridae* (Fig 1A).

### Mitovirus genome

We assembled a 2,372 bp viral genome (Fig 1B) from the metatranscriptome data. There was a single blast hit when we used Blastn to search the nt database, a mitoviral genome from a soil metatranscriptome (ON163930.1). We used the yeast mitochondrial translation table to identify a 647 aa protein translated from ORF 3. An InterPro search of the amino acid sequence identified a "mitoviral RNA-dependent RNA polymerase" (IPR008686). A Blastp search of the amino acid sequence against the nr database produced hits to various *Mitovirus* genomes that infect fungi and plants, with the highest percent homology being a 51% match with "Hangzhou mitovirus 7" (UHK03006.1), which was assembled from a metatranscriptome of a dragonfly. Blastp results also included hits to *Entomophthora muscae mitovirus 1* (35% identity) and 2

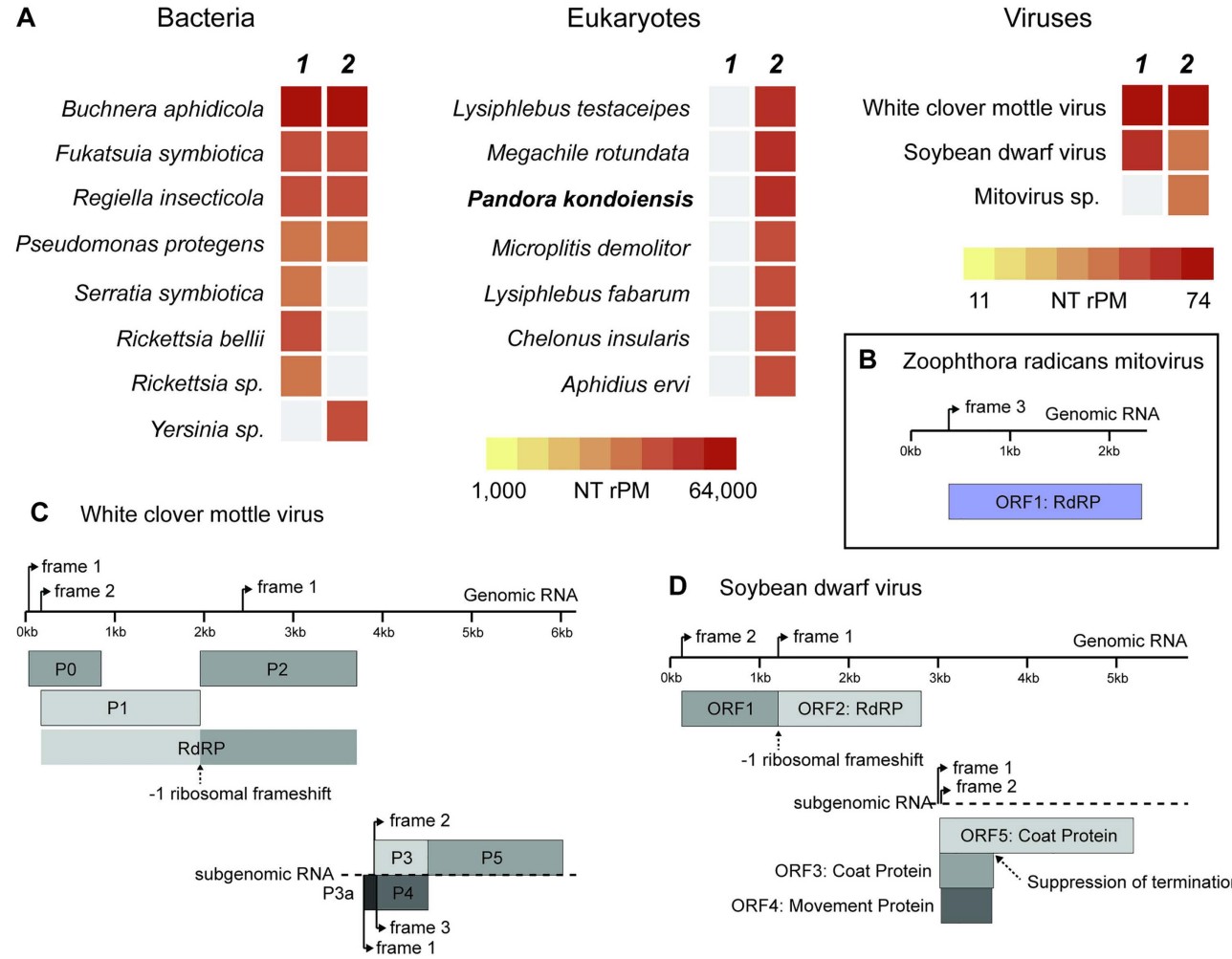

**Fig 1. Metatranscriptome analysis and viral genome assembly. A** shows heatmaps with the taxonomic assignment of reads in the metatranscriptome dataset, with darker red colors indicating a higher number of reads per million (RPM). Bacteria are shown to the left, eukaryotes in the middle, and viruses to the right of the figure. The two metatranscriptome libraries (pools 1 and 2) are shown as indicated. **B** is a visual depiction of the *Mitovirus* genome, showing the RNA genome and the single ORF that encodes for an RdRP. **C** and **D** are visual depictions of the two plant viral genomes (*White clover mottle virus*, left; *Soybean dwarf virus*, right). In both figures, genomic RNA is shown along the top of the figures, from which an RdRP is encoded over two ORFs and a −1 ribosomal frameshift. The sub-genomic RNAs are shown at the bottom of the panels, which encode for capsid and transmission proteins as indicated.

(32% identity), which are recently described mitoviruses isolated from an insect entomopathogen [16]. We deposited the sequence for this genome in NCBI (GenBank: PQ606655.1).

## Polerovirus genome

We assembled a nearly complete genome for a *Polerovirus* from Pool 1 (NCBI accession: PV560925), which we identified as *White clover mottle virus* (WCMV, *Solemoviridae*). Little is known about this virus apart from a published sequence for its genome (NC_031747.1) showing it consists of a single, positive-strand RNA genome (Fig 1C). Our 6,170 bp genome is 98.4% identical to this reference strain. We assembled a second 6,166 bp genome for this virus from pool 2 (NCBI cession PQ553207.1), which was 99.1% identical to the pool 1 genome sequence, suggesting there may be multiple strains

of the virus circulating within aphid populations. In addition to homologous protein sequences for those in the published reference genome, we included a protein sequence for P3a based on our sequence data similar to other poleroviruses (Fig 1C). Both pools had relatively high and even coverage across the genome (74.4 and 73.0 rPM in pools 1 and 2, respectively).

### Luteovirus genome

We assembled a 5,802 bp genome identified as *Soybean Dwarf Virus* (SbDV, *Tombusviridae*) from RNAseq reads from pool 1 (NCBI accession: PV611337). This is a *Luteovirus* that is emerging in the United States with potential economic impact, and is transmitted by multiple aphid vectors (e.g. *Aulacorthum solani* and *Acyrthosiphon pisum*) in a persistent non-propagative manner [28]. There were substantially fewer reads from this virus in pool 2 and a full genome could not be assembled. We identified coding sequences for coat and movement proteins and for the RdRP based on a fusion of two open reading frames and a ribosomal frameshift (Fig 1D), as in another luteoviruses. At the nucleotide level, our genome is 95.6% similar to the closest SBDV isolate (MIR20SW; accession OL472235.1). The RdRP amino acid sequence was 98.99% similar to this SbDV isolate.

### Field *Mitovirus* screening

Because mitoviruses are typically associated with fungi, and because we found *Mitovirus* reads only in the metatranscriptome pool that also had reads with homology to fungal pathogens, we hypothesized that this virus might have a fungal rather than aphid host. To investigate this further, we developed screening primers for the *Mitovirus*, and screened 19 field samples, including two visible 'fungal cadavers', i.e., samples that had become infected with a fungal entomopathogen and then developed a visible sporulating cadaver when brought into the lab. Two samples tested positive among these samples, but the fungal cadavers tested negative for *Mitovirus* (S1 Fig).

### 18S amplicon sequencing

We carried out 18S sequencing of 19 field samples, to identify fungal pathogens circulating in the local aphid population and determine if the *Mitovirus*-positive samples were also associated with a particular species of fungus. Both *Mitovirus*-positive samples had a high relative abundance (99% of taxonomically classified amplicons) of reads from *Zoophthora radicans,* a fungal entomopathogen. *Z. radicans* was not found in any of the *Mitovirus*-negative samples (Fig 2). The *Z. radicans-Mitovirus* association was statistically significant (Fisher's exact test; p = 0.0058). The two fungal cadavers and two additional samples were identified with reads from *Pandora neoaphidis*. None of the four samples with *Pandora* infections tested positive for *Mitovirus* by PCR. Species demarcation in this study was based on ICTV *Mitoviridae* Study Group (SG) criteria of less than 70% amino acid sequence identity to other viral genome sequences [29]. *Zoopthora radicans mitovirus 1* exhibits less than 70% amino acid sequence identity to the best match in the nr database (see above). We, therefore, concluded that this sequence is a newly described species of *Mitovirus* that is likely associated with *Z. radicans*, and in the NCBI submission we refer to this virus as *Zoophthora radicans mitovirus 1*.

## Discussion

This study illustrates the utility of next-generation sequencing for uncovering novel microbial diversity in natural populations, even in well-studied organisms like pea aphids. Using ribosomal RNA-depleted metatranscriptome sequencing, we discovered a new species of virus in the family *Mitoviridae.* We screened a population of aphids for the newly described *Mitovirus* and then performed 18S amplicon sequencing to characterize the fungal community associated with pea aphids. We used these data sets to infer that our newly described *Mitovirus* is likely associated with an entomopathogenic fungus called *Zoophthora radicans*. We note, however, that the true host of this novel mitovirus cannot be definitely established

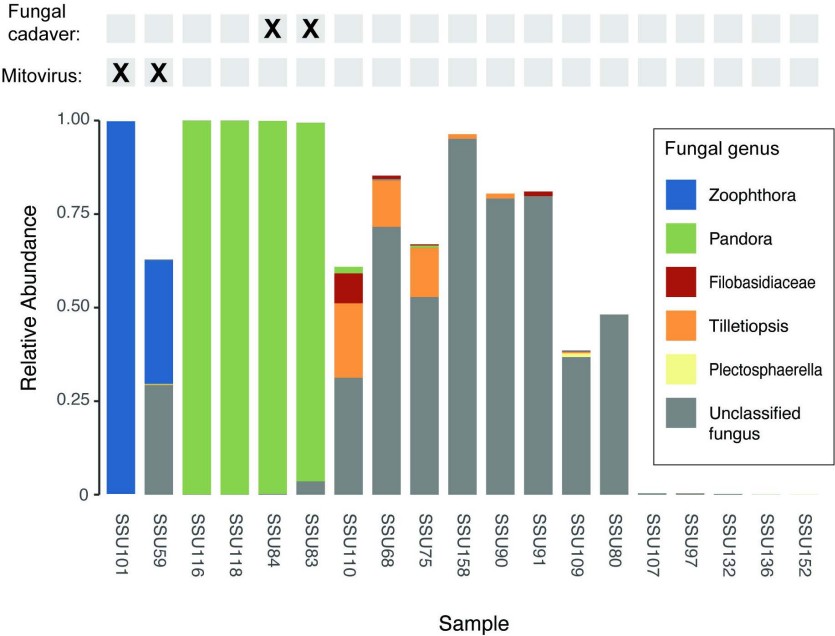

**Fig 2. Relative abundance of fungi, identified via 18s rRNA amplicon sequencing.** The Y-axis shows the relative abundance of fungal taxa, and each sample is shown along the X-axis. The white space is representative of the relative abundance of amplicons assigned to non-fungal taxonomy. The top of the figure shows the results of PCR screens of each sample for *Mitovirus* and whether each aphid sample had formed a visible entomopathogen cadaver.

without in vitro isolation and culturing, and caution is warranted in the use of the name *Zoophthora radicans mitovirus 1* without further investigation.

*Mitoviruses* typically infect and replicate within the mitochondria of fungi, although they can occasionally migrate into the fungal cytoplasm [15]. *Mitoviruses* have been investigated for their potential as biocontrol agents against plant pathogenic fungi [30] as they are associated with hypovirulence, abnormal mitochondrial morphology, and reduced in vitro growth in fungi such as *Botrytis cinerea* and *Sclerotinia sclerotiorum* [31–34]. These studies suggest that mitoviruses can be transmitted horizontally between fungal strains via hyphal anastomosis (fusion of multiple hyphae), though it remains unclear whether this involves the sharing of infected mitochondria or cytoplasm. *Mitovirus* infections can also persist through vertical transmission, including via fungal spores [16]. While the precise role of *Zoophthora radicans mitovirus 1* remains speculative, it is possible that it influences the fitness of its host similarly to how other entomopathogens use viruses to modulate host interactions and virulence [35].

Several viruses are known to be associated with *A. pisum*, including *Acyrthosiphon pisum virus* (APV), a Picorna-like virus found independently in two laboratory cultures [9,11]. However, little is known about the ecology of pea aphid-specific viruses in natural populations. In our study, we found no evidence of infection with any insect-specific (i.e., non-vectored) pathogens in the field-collected aphids. To gain a more comprehensive understanding of viral diversity in this important model organism, future work should include metatranscriptome sequencing of *A. pisum* from diverse geographical locations and a range of host plants. We did recover nearly complete genomes of *White clover mottle virus* (WCMV) from two pools of aphids. High coverage suggests that aphids may act as a vector for this virus, but additional data is needed to confirm a role for aphids in viral transmission. Alternatively, the presence of WCMV sequences could reflect ingestion of virus-containing plant material rather than a true vector association with the aphid.

Our study underscores the importance of molecular approaches for characterizing microbial communities in field-collected specimens. This aphid population in Knoxville, TN, has been under investigation in our lab for several years. We have identified multiple entomopathogenic fungi, including *Batkoa major*, *Pandora neoaphidis*, and *Conidiobolus spp.*, but until now were unaware of the presence of *Zoophthora radicans* this population. The failure to identify *Zoophthora* is likely due to limitations in traditional spore morphology-based identification, which remains the classical method for diagnosing entomopathogens. Notably, although *Z. radicans* was found infecting the field-collected aphids at a similar relative abundance to *P. neoaphidis* aphid cadavers, the *Z. radicans*-infected aphids did not develop cadavers, which could be relevant to the infection dynamics of these fungi in the field. Our study highlights the critical role of molecular tools in fungal identification, as traditional methods may miss key pathogens in natural populations, particularly in generalist species like *Z. radicans*, where spore morphology can be indistinguishable from other fungi.

## Supporting information

**S1 Table. Individual aphid metadata.** Host plant was identified to the genus and when confident, to species level.
(XLSX)

**S1 Fig. Gel image of *Mitovirus* PCR screening of field-collected aphids.** Gel image was taken via iPhone camera from UV transilluminator, inverted and labelled using Biorender.
(TIF)

## Author contributions

**Conceptualization:** Benjamin J. Parker.

**Data curation:** Paula Rozo-Lopez, Benjamin J. Parker.

**Formal analysis:** Meaghan J. Adler, Paula Rozo-Lopez, Benjamin J. Parker.

**Funding acquisition:** Benjamin J. Parker.

**Investigation:** Meaghan J. Adler, McKayla M. Martin, Paula Rozo-Lopez, Benjamin J. Parker.

**Methodology:** Meaghan J. Adler, McKayla M. Martin, Paula Rozo-Lopez, Benjamin J. Parker.

**Supervision:** Benjamin J. Parker.

**Visualization:** Meaghan J. Adler, Benjamin J. Parker.

**Writing – original draft:** Meaghan J. Adler, Benjamin J. Parker.

**Writing – review & editing:** Meaghan J. Adler, Paula Rozo-Lopez, Benjamin J. Parker.

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
