## [Decision Letter · Decision Letter 0]

1 Jul 2025

PONE-D-25-25639A novel mitovirus associated with the fungal entomopathogen *Zoophthora radicans*PLOS ONE

Dear Dr. Adler,

Thank you for submitting your manuscript to PLOS ONE. After careful consideration, we feel that it has merit but does not fully meet PLOS ONE’s publication criteria as it currently stands. Therefore, we invite you to submit a revised version of the manuscript that addresses the points raised during the review process.

Please revise and improve as mentioned by reviewer #2 data availability, methods descriptions, and phylogenetic relationships, as their suggestions are relevant for reproducibility and transparency. I do not expect a virus isolation experiment, just keep in mind the comments to revise in the discussion the limitations for host assignment.

We look forward to receiving your revised manuscript.

Kind regards,

Humberto Julio Debat

Academic Editor

PLOS ONE

Journal Requirements: 

 [This work was funded by National Science Foundation (NSF) Grant IOS‐2152954 to BJP and by Grant DEB‐2305653 to PRL. MJA was supported by the SMART scholarship funded by OUSD/R&E (The Under Secretary of Defense-Research and Engineering), National Defense Education Program (NDEP) / BA-1, Basic Research. MMM was supported by a Microbiology Summer Research Award from the Department of Microbiology at UTK. A portion of this project was carried out by MMM in association with a course-based undergraduate research experience (CURE), which was funded by the College of Arts and Sciences at the University of Tennessee Knoxville. BJP is a Pew Scholar in the Biomedical Sciences, funded by the Pew Charitable Trusts.]. 

[MJA was supported by the SMART scholarship funded by OUSD/R&E (The Under Secretary of Defense-Research and Engineering), National Defense Education Program (NDEP) / BA-1, Basic Research. The project was supported by National Science Foundation (NSF) Grant IOS‐2152954 to BJP and by Grant DEB‐2305653 to PRL. MMM was supported by a Microbiology Summer Research Award from the Department of Microbiology at UTK. A portion of this project was carried out by MMM in association with a course-based undergraduate research experience (CURE), which was funded by the College of Arts and Sciences at the University of Tennessee Knoxville. BJP is a Pew Scholar in the Biomedical Sciences, funded by the Pew Charitable Trusts.]

[This work was funded by National Science Foundation (NSF) Grant IOS‐2152954 to BJP and by Grant DEB‐2305653 to PRL. MJA was supported by the SMART scholarship funded by OUSD/R&E (The Under Secretary of Defense-Research and Engineering), National Defense Education Program (NDEP) / BA-1, Basic Research. MMM was supported by a Microbiology Summer Research Award from the Department of Microbiology at UTK. A portion of this project was carried out by MMM in association with a course-based undergraduate research experience (CURE), which was funded by the College of Arts and Sciences at the University of Tennessee Knoxville. BJP is a Pew Scholar in the Biomedical Sciences, funded by the Pew Charitable Trusts.]. 

Additional Editor Comments:

Please revise and improve as mentioned by reviewer #2 data availability, methods descriptions, and phylogenetic relationships, as their suggestions are relevant for reproducibility and transparency. I do not expect a virus isolation experiment, just keep in mind the comments to revise in the discussion the limitations for host assignment.

Reviewers' comments:

Reviewer's Responses to Questions

**Comments to the Author**

1. Is the manuscript technically sound, and do the data support the conclusions?

Reviewer #1: Yes

Reviewer #2: Partly

2. Has the statistical analysis been performed appropriately and rigorously? 

Reviewer #1: Yes

Reviewer #2: Yes

3. Have the authors made all data underlying the findings in their manuscript fully available?

Reviewer #1: Yes

Reviewer #2: No

4. Is the manuscript presented in an intelligible fashion and written in standard English?

Reviewer #1: Yes

Reviewer #2: Yes

5. Review Comments to the Author

Reviewer #1: The manuscript entitled “A novel mitovirus associated with the fungal entomopathogen Zoophthora radicans” by Adler et al. is a straightforward manuscript that reports the molecular characterization of a novel mitovirus associated to a pea aphid´s fungal entomopathogen. It is great that authors conducted the 18S amplicon sequencing and further PCR screening to unequivocally assign the host to the identified mitovirus.

Reviewer #2: The study by Adler et al. identifies several viruses present in metatranscriptomic sequencing data for field-collected aphids. Of these viruses, the researchers recovered a contig encoding a novel mitovirus, and its presence in aphids was associated with the co-presence of reads attributed to the fungal entomopathogen, Zoophthora radicans. The data provided is original and provides novel insights into aphid virome diversity, which also includes viruses from plants and fungi. However, there are several minor revisions regarding data availability and methods descriptions, as well major revisions regarding the phylogenetic relationships and the true host of “Zoophthora radicans mitovirus”:

- Individual aphid sample metadata should be provided either in main figures or supplemental table:

o Which samples were from winged (alate) versus unwinged (aptera) aphids?

o Which samples were aphids from T. repens versus T. pratense?

o Which individual aphid samples were put into pool 1 and pool 2 used for metatranscriptomic sequencing?

- For data reproducibility purposes, the method of cDNA synthesis used for PCR screening individual aphid samples for the novel mitovirus should be stated; it is unclear what “constructed cDNA as above” (line 127) is referring to.

- The data provided for RT-PCR confirmation of the mitovirus across the 10 aphid samples is insufficient in Figure 2. PCR gel images should be shown including expected/observed molecular weight of mitovirus amplicons as well as appropriate control groups, such as reference gene controls, cDNA synthesis reactions plus and minus reverse transcriptase, and/or mitovirus RT-PCR for the corresponding gDNA samples.

- A statement of what criteria was chosen to call this a “novel” species of Mitoviridae should be included. For example, refer to the International Committee on Taxonomy of Viruses (ICTV) Mitoviridae proposal for this criteria (https://ictv.global/taxonomy). A phylogenetic tree comparing the novel mitovirus RdRp to several other known mitovirus RdRps would also provide a Mitoviridae genus-level assignment and provide a measure of novelty of Zoophthora radicans mitovirus compared to other mitoviruses

- For data reproducibility purposes, statements for the parameters used with following tools should be included, even if simply “default parameters were used”:

o STAR

o Trimmomatic

o PriceSeqFilter

o Mini-map2

o Diamond

o SPADES

o bowtie2

o Geneious Prime

o NCBI ORFfinder

o InterPro

- The “true” host of the novel mitovirus could be established by in vitro isolation and culture of Z. radicans from a mitovirus-positive aphid, then showing that Z. radicans alone is RT-PCR positive for the mitovirus. This experiment should be performed if it’s trivial, as this would confirm the host of the novel mitovirus (otherwise it should not be called “Zoophthora radicans mitovirus”).

6. PLOS authors have the option to publish the peer review history of their article (what does this mean? ). If published, this will include your full peer review and any attached files.

**Do you want your identity to be public for this peer review?** For information about this choice, including consent withdrawal, please see our Privacy Policy .

Reviewer #1: **Yes: ** Nicolas Bejerman

Reviewer #2: No

---

## [Author Response · Author response to Decision Letter 1]

7 Aug 2025

Reponses to reviewers for PLOS One

Additional Editor Comments: Please revise and improve as mentioned by reviewer #2 data availability, methods descriptions, and phylogenetic relationships, as their suggestions are relevant for reproducibility and transparency. I do not expect a virus isolation experiment, just keep in mind the comments to revise in the discussion the limitations for host assignment.

Thanks to the editor for handling this manuscript.

Reviewer #1: The manuscript entitled “A novel mitovirus associated with the fungal entomopathogen Zoophthora radicans” by Adler et al. is a straightforward manuscript that reports the molecular characterization of a novel mitovirus associated to a pea aphid´s fungal entomopathogen. It is great that authors conducted the 18S amplicon sequencing and further PCR screening to unequivocally assign the host to the identified mitovirus.

Thanks to Reviewer #1 for their review of this manuscript.

Reviewer #2: The study by Adler et al. identifies several viruses present in metatranscriptomic sequencing data for field-collected aphids. Of these viruses, the researchers recovered a contig encoding a novel mitovirus, and its presence in aphids was associated with the co-presence of reads attributed to the fungal entomopathogen, Zoophthora radicans. The data provided is original and provides novel insights into aphid virome diversity, which also includes viruses from plants and fungi. However, there are several minor revisions regarding data availability and methods descriptions, as well major revisions regarding the phylogenetic relationships and the true host of “Zoophthora radicans mitovirus”:

Thanks to Reviewer #2 for their comments on the manuscript.

Comment #1:

- Individual aphid sample metadata should be provided either in main figures or supplemental table:

• Which samples were from winged (alate) versus unwinged (aptera) aphids?

• Which samples were aphids from T. repens versus T. pratense?

• Which individual aphid samples were put into pool 1 and pool 2 used for metatranscriptomic sequencing?

As suggested, we now include a table in the Supporting Information (S1 Table) that describes this metadata. Readers are directed to S1 Table in the following methods section sentence: “Samples were a mix of winged (alate) and unwinged (aptera) adults, metadata in S1 Table.” Host plants were identified at a genus level and to a species level, when confident.

Comment #2

- For data reproducibility purposes, the method of cDNA synthesis used for PCR screening individual aphid samples for the novel mitovirus should be stated; it is unclear what “constructed cDNA as above” (line 127) is referring to.

We added the specific cDNA synthesis kit to the referenced sentence. It now reads, “We extracted RNA from field-collected aphids and constructed cDNA via iScript cDNA Synthesis Kit by BIO-RAD, as directed by the manufacturer.”

Comment #3

- The data provided for RT-PCR confirmation of the mitovirus across the 10 aphid samples is insufficient in Figure 2. PCR gel images should be shown including expected/observed molecular weight of mitovirus amplicons as well as appropriate control groups, such as reference gene controls, cDNA synthesis reactions plus and minus reverse transcriptase, and/or mitovirus RT-PCR for the corresponding gDNA samples.

We have added a supporting figure, S2 Figure, which is gel images of the Mitovirus PCR screening of aphid samples used in amplicon sequencing in Figure 2.

Comment #4

- A statement of what criteria was chosen to call this a “novel” species of Mitoviridae should be included. For example, refer to the International Committee on Taxonomy of Viruses (ICTV) Mitoviridae proposal for this criteria (https://ictv.global/taxonomy). A phylogenetic tree comparing the novel mitovirus RdRp to several other known mitovirus RdRps would also provide a Mitoviridae genus-level assignment and provide a measure of novelty of Zoophthora radicans mitovirus compared to other mitoviruses

In the revised results we now clarify: “Species demarcation in this study was based on ICTV Mitoviridae Study Group (SG) criteria of less than 70% amino acid sequence identity to other viral genome sequences (Walker, et al. 2022). Zoopthora radicans mitovirus 1 exhibits less than 70% amino acid sequence identity to the best match in the nr database (see above). We, therefore, concluded that this sequence is a newly described species of Mitovirus that is likely associated with Z. radicans, and in the NCBI submission we refer to this virus as Zoophthora radicans mitovirus 1.”

Comment #5

- For data reproducibility purposes, statements for the parameters used with following tools should be included, even if simply “default parameters were used”:

o STAR

o Trimmomatic

o PriceSeqFilter

o Mini-map2

o Diamond

o SPADES

o bowtie2

o Geneious Prime

o NCBI ORFfinder

o InterPro

We have added specific parameters to the bioinformatic processing methods section. The CZ ID Illumina mNGS metagenomic pipeline is publicly available on their github so we have clarified when the pipeline starts and ends in our analysis. The remaining programs (Geneious Prime, NCBI ORFfinder, InterPro) are now described with the version used and the description “with default parameters” as suggested by Reviewer #2 version. It now reads, “With completion of the CZ ID pipeline, we de novo assembled reads with homology to each putative virus using Geneious Prime version 2025.0 using default parameters for the De Novo Assemble tool. With the assembled genomes, we used the NCBI Open Reading Frame Finder (https://www.ncbi.nlm.nih.gov/orffinder/, last modified 2021-05-04) to identify protein-coding regions of the viral genome (using genetic code 1 for putative viruses, and genetic code 3 Yeast Mitochondrial for the putative mitovirus). We used interpro (https://www.ebi.ac.uk/interpro/ release 103.0) to look for conserved genes to annotate viral genomes.”

Comment #6

- The “true” host of the novel mitovirus could be established by in vitro isolation and culture of Z. radicans from a mitovirus-positive aphid, then showing that Z. radicans alone is RT-PCR positive for the mitovirus. This experiment should be performed if it’s trivial, as this would confirm the host of the novel mitovirus (otherwise it should not be called “Zoophthora radicans mitovirus”).

This would be a definitive way to associate the mitovirus to Z.radicans, but unfortunately, neither in vitro isolation nor culture of a Z.radicans from a known mitovirus-positive field collected aphid is possible. We have used the following sentence to describe the inference made from our data, “We used these data sets to infer that our newly described Mitovirus is likely associated with an entomopathogenic fungus called Zoophthora radicans”. Further, we add an additional caveat in the discussion: “We note, however, that the true host of this novel mitovirus cannot be definitely established without in vitro isolation and culturing, and caution is warranted in the use of the name Zoophthora radicans mitovirus 1 without further investigation.”

---

## [Editor Report · Decision Letter 1]

13 Aug 2025

A novel mitovirus associated with the fungal entomopathogen *Zoophthora radicans*

PONE-D-25-25639R1

Dear Dr. Adler,

We’re pleased to inform you that your manuscript has been judged scientifically suitable for publication and will be formally accepted for publication once it meets all outstanding technical requirements.

Kind regards,

Humberto Julio Debat

Academic Editor

PLOS ONE
---

## [Editor Report · Acceptance letter]

PONE-D-25-25639R1

PLOS ONE

Dear Dr. Adler,

I'm pleased to inform you that your manuscript has been deemed suitable for publication in PLOS ONE. Congratulations! Your manuscript is now being handed over to our production team.

Kind regards,

on behalf of

Professor Humberto Julio Debat

Academic Editor

PLOS ONE